# The Use of Host Biomarkers for the Management of Invasive Fungal Disease

**DOI:** 10.3390/jof8121307

**Published:** 2022-12-16

**Authors:** James S. Griffiths, Selinda J. Orr, Charles Oliver Morton, Juergen Loeffler, P. Lewis White

**Affiliations:** 1Centre for Host-Microbiome Interactions, Faculty of Dentistry, Oral and Craniofacial Sciences, King’s College London, London WC2R 2LS, UK; 2Wellcome-Wolfson Institute for Experimental Medicine, School of Medicine, Dentistry and Biomedical Science, Queen’s University Belfast, Belfast BT9 7BL, UK; 3School of Science, Western Sydney University, Campbelltown, NSW 2751, Australia; 4Department of Internal Medicine II, University Hospital of Würzburg, 97070 Würzburg, Germany; 5Public Health Wales, Microbiology Cardiff, University Hospital of Wales, Heath Park, Cardiff CF14 4XW, UK

**Keywords:** invasive fungal disease, host biomarkers, genomics, transcriptomics, proteomics, fungal diagnostics

## Abstract

Invasive fungal disease (IFD) causes severe morbidity and mortality, and the number of IFD cases is increasing. Exposure to opportunistic fungal pathogens is inevitable, but not all patients with underlying diseases increasing susceptibility to IFD, develop it. IFD diagnosis currently uses fungal biomarkers and clinical risk/presentation to stratify high-risk patients and classifies them into possible, probable, and proven IFD. However, the fungal species responsible for IFD are highly diverse and present numerous diagnostic challenges, which culminates in the empirical anti-fungal treatment of patients at risk of IFD. Recent studies have focussed on host-derived biomarkers that may mediate IFD risk and can be used to predict, and even identify IFD. The identification of novel host genetic variants, host gene expression changes, and host protein expression (cytokines and chemokines) associated with increased risk of IFD has enhanced our understanding of why only some patients at risk of IFD actually develop disease. Furthermore, these host biomarkers when incorporated into predictive models alongside conventional diagnostic techniques enhance predictive and diagnostic results. Once validated in larger studies, host biomarkers associated with IFD may optimize the clinical management of populations at risk of IFD. This review will summarise the latest developments in the identification of host biomarkers for IFD, their use in predictive modelling and their potential application/usefulness for informing clinical decisions.

## 1. Background

Opportunistic invasive fungal diseases (IFD) cause severe disease morbidity and mortality globally, with case numbers increasing in line with an expanding susceptible population. Recent estimates have put IFD mortality at more than 1.5 million people per year, comparable to the yearly mortality resulting from tuberculosis or HIV and malaria combined [1,2,3,4]. Whilst exposure to fungi that cause IFD is inevitable, and an estimated 15% of the population experience a mild, non-invasive fungal infection each year, generally only those with significant clinical intervention or a defective/compromised immune system develop invasive disease. The extensive use of immune suppressing or modulating therapies is only increasing, with the corresponding incidence of IFD likely to follow suit. Given the inevitable exposure to commensal or environmental fungi, there is considerable concern regarding IFD in the populations at risk, managed typically through prophylactic or empirical antifungal strategies. However, not every patient develops IFD despite similar clinical risk within at-risk cohorts and unavoidable fungal exposure. Therefore, it is likely host factors (e.g., genetics and subsequent immunity) specific to each individual may play a role in IFD susceptibility. This review will initially discuss the major contributors to IFD, their clinical importance and current diagnostic strategies with a focus on limitations. We will then investigate the potential application of new host biomarkers to aid in the management of IFD, through a summary review of publications describing research only validated in clinical cohorts. These novel risk factors, together with an understanding of existing and novel clinical risk factors, and advances in fungal diagnostics, may promote the rapid, accurate, and individualized diagnosis of IFD in the future.

## 2. An Overview of Invasive Fungal Disease

Numerous fungal genera and species cause a wide range of IFD. This highly diverse array of fungi represents a huge challenge for the accurate and rapid diagnosis and targeted treatment of IFD. This is complicated further by the patient’s primary disease, immune status, and clinical presentation. Two fungal species, *Candida* and *Aspergillus*, are responsible for the vast majority of IFD. However, other fungi such as *Zygomycetes* (now the *Mucoromycota* and *Zoopagomycota*), and fungal species such as *Pneumocystis* and *Cryptococcus*, significantly contribute to IFD incidence [5].

## 3. Invasive Aspergillosis

Invasive aspergillosis (IA), caused by fungi in the genus *Aspergillus*, has become a leading cause of death in immune compromised patient cohorts. *Aspergillus fumigatus* is primarily responsible for IA but other species including *Aspergillus flavus*, *Aspergillus niger*, *Aspergillus terreus* and *Aspergillus nidulans* are also responsible for human disease, sub-species with the *A. fumigati* complex are difficult to differentiate from *A. fumigatus* but may have markedly different antifungal susceptibility profiles [6,7,8,9]. *Aspergillus* is highly prevalent in the environment and it is estimated humans inhale hundreds of *Aspergillus* conidia each day [10]. In healthy individuals, inhaled *Aspergillus* conidia are rapidly cleared by the immune system. Problems arise when conidia are not cleared and persisting conidia germinate into hyphae within the lung and cause local infection, inflammation, necrosis, and fibrosis [11]. The clinical need for improvements in IA diagnosis is clear. *Aspergillus*, if poorly controlled in the lung, can disseminate via blood vessels throughout the body; once systemic and/or cerebral, IA mortality rates approach 100% [10,12,13]. Annual estimates predict 300,000 cases of IA, primarily in patients with haematologic malignancies with high (40%) mortality rates [14,15,16]. The almost constant environmental exposure to *Aspergillus* spp. makes diagnosing IA particularly challenging and requires strategies involving IA-specific biomarkers. Currently, a range of clinical signs and mycological tests are used to diagnose IA, but combined diagnostics (e.g., *Aspergillus* antigen and PCR) appear optimal, ideally suited for excluding infection. Current diagnostic investigations and clinical intervention guidelines (as defined by the EORTC) are limited by their inability to discriminate between different *Aspergillus* spp. with diagnoses regular achieved in the absence of culture or species level identification [17]. The performance of assays can vary dependent on the species, with many assays designed/optimized for the detection of the most prevalent species (i.e., *A. fumigatus*) [18].

Recently, IA has been identified in patients secondary to respiratory viral infection; influenza-associated pulmonary aspergillosis (IAPA) and COVID-19-associated pulmonary aspergillosis (CAPA) are frequently diagnosed in intensive care unit patients, potentially doubling annual estimates of IA [19,20]. *Aspergillus* infection, persistence and subsequent invasion into tissues is not homogenous between patients and presentation is heavily influenced by patient’s primary disease and immune state; IA diagnosis is complex and challenging. 

## 4. Invasive Candidiasis

*Candida* spp. are commensal organisms found throughout the human body on mucosal membranes of gastro-intestinal, oral, respiratory, and genito-urinary tracts. Invasive candidiasis (IC) arises following microbiological imbalance, anatomical disruption or immune intervention and can be broadly classed into candidemia and deep-seated candidiasis and both conditions can present separately or in combination. IC usually results from commensal *Candida* yeast populations overgrowing their niche, releasing virulence factors, and penetrating through mucosal barrier sites [21,22]. However, infection may also arise when *Candida* is introduced into sterile tissues through surgery or via a catheter [23]. Annually there are an estimated 500,000 cases of IC, primarily caused by *Candida albicans*, *Candida glabrata*, *Candida tropicalis*, *Candida parapsilosis* and *Candida krusei* [24,25,26]. The diagnosis of IC is complicated by the range of *Candida* species that can cause invasive disease, the variety of invasive disease manifestations, and by *Candida*’s commensal status. Accordingly, mycology tests may not provide sensitive and accurate results even when testing sterile sites and species level identification is required to target therapy appropriately. Like IA diagnosis, a combination of clinical and mycological biomarkers are currently used to diagnose IC, but an optimal strategy is yet to be determined.

## 5. IFD—General Considerations

A range of other fungal genera are associated with IFD, with *Cryptococcus* spp. and *Pneumocystis jirovecii* associated with severe disease burden. Together with IA and IC, these fungal pathogens are responsible for 90% of fungal-associated mortality [27,28]. However, unlike IA and IC the diagnosis of cryptococcosis and *Pneumocystis* pneumonia is achievable through fungal diagnostic testing (i.e., *Cryptococcus* antigen testing or *Pneumocystis* PCR) [29,30]. Other fungal infections (e.g., Mucormycosis, Scedosporiosis, or Fusariosis) while increasingly being diagnosed, currently have limited or no reports on host response that could be used as a potential host biomarker specific to that infection [31,32,33].

These main causative agents of IFD represent a huge diagnostic challenge. Clinically relevant fungal pathogens may be commensal or environmentally acquired, progress through multiple morphologies, each with their own pathogen associated molecular patterns and virulence factors, infect a variety of tissues, and cause a range of disease from mild superficial infections through to life-threatening invasive disease. To complicate matters further, a wide and diverse array of patient cohorts are susceptible to IFD, typically defined by some form of compromised immune system or clinical intervention. Crucially, whilst the IFD experienced by these patient cohorts may ultimately lead to disseminated fungal infection, the path to get there is highly disparate. This makes diagnosing IFD particularly challenging and a one-size fits all approach may not be appropriate for patients.

## 6. Current Fungal Diagnosis and Therapeutics

There are two defining features of IFD: (i) fungi are unavoidable; and (ii) a significant clinical risk/anatomical change or immunological defect is required for pathogenic fungal growth and the development of IFD. Given the serious threat IFD poses to patients and the significant clinical burden associated with IFD, effective therapeutics, and rapid and accurate diagnosis are essential. Fungi present several challenges to both aspects.

The diagnosis of fungal infections in the clinic involves assessment of the patient’s underlying risk of IFD, focussed on the individual’s immune status and underlying disease. Here, patients are stratified between low and high-risk groups for specific IFD. Beyond the prediction of clinical risk, patients are further stratified into proven, probable, and possible IFD based on histological confirmation of pathology or the combination of clinical/radiological evidence and mycology [34]. In addition to conventional mycology, detection of fungal antigens (galactomannan, mannan and (1-3)-β-D-Glucan), fungal nucleic acid using molecular tests and radiological imaging are typically used to identify IFD. However, these diagnostic tests have limitations, primarily that a positive result is not always indicative of IFD [35]. Individual test performance is variable, dependent on IFD, but molecular testing appears promising for the detection of IC and *Pneumocystis* pneumonia, and cryptoccocal antigen testing is excellent for the diagnosis of cryptococcosis (Table 1). Both biomarker and molecular testing are far from perfect for the diagnosis of IA when used alone, and combination testing is required to overcome performance limitations [35].

The incidence of IFD even in populations at risk remains modest and the development of combined biomarker diagnostic driven approaches for managing IFD have proven useful by excluding disease when tests are consistently negative (NPV > 99%), alleviating the need for unnecessary antifungal therapy [57]. Conversely, even when multiple tests are positive the probability of IFD is far from certain (approximately 50%) and proven IFD, dependent on culture or histology from a sterile site is rarely achieved ante-mortem [35]. Regularly, healthcare professionals manage patients with probable or possible IFD, where patients present with clinical manifestations typical of certain infections (e.g., for IA: halos, nodules cavities on chest CT) with or without mycological evidence, respectively [34]. For cases of probable IFD, the diagnosis and subsequent treatment can be confidently administered as the probability of disease is high. For cases of possible IFD, lacking mycological evidence the case is less convincing, the typical signs could be associated with other infections or clinical conditions, and given that combined biomarker negativity is sufficient to exclude IFD there is an argument that cases of possible IFD should be downgraded when mycological investigations are consistently negative [33]. In the clinic, where patients at risk of IFD present with non-specific signs of infection and have significant mycological evidence of IFD, yet lack typical clinical signs of IFD, pre-emptive antifungal therapy may be administered or a diagnostic work-up initiated in an attempt to gain sufficient clinical evidence to categorise the certainty of infection in the currently undefined patient. In all these scenarios, the diagnosis of IFD may be enhanced by looking for host biomarkers via genomic, transcriptomic, and proteomic investigation to identify mutations, RNA transcriptome profiles, peripheral blood mononuclear cell or whole blood cell response or immune cytokine/chemokine levels that may be indicative of increased risk of IFD, or even representative of current IFD. While combined fungal biomarker strategies have demonstrated utility for excluding IFD and subsequent need to treat, their use is not widespread, a limitation that can also be applied to the histological investigation of tissue biopsies when trying to confirm IFD [32,33]. The result of this limited diagnostic capability, coupled with poor treatment outcomes, is the empirical treatment of patients [34]. Based primarily on treating underlying clinical risk or upon the first indication of potential IFD (usually refractory fever), less targeted treatment is ill-advised in an era of increasing antifungal resistance [35,58].

Alongside diagnostic challenges, anti-fungal therapeutics are encountering issues. Currently treatment for IFD relies on four classes of antifungal therapeutics: azoles, polyenes, echinocandins and the pyrimidine analogue 5-flucytosine. However, treatment can be associated with toxicity, side-effects and drug-drug interactions and fungi have adapted numerous mechanisms to avoid their action [59]. Antifungal resistance can be driven by environmental and agricultural practices, and is associated with resistance in clinical settings, including multidrug resistant species [60,61,62,63]. Evidence also suggests therapeutics have become less effective against established infections, highlighting the need for early diagnosis or pre-emptive strategies [35,57]. Additionally, the administration of antifungal therapies may require modulation of patient’s primary care, further emphasising the requirement for accurate and targeted diagnosis [64]. A balance must be struck between patient’s primary therapy and resulting immune suppression, and their fungal susceptibility. In an emerging scenario of increasing antifungal resistance and limited drug classes, it is critical that we develop fungal diagnostics to accurately target antifungal therapy to the patients who need it and exclude the need for antifungal therapy in patients with false positive results associated with current diagnostics or where there is clinical pressure to administer empirical therapy.

Whilst there is a need for novel assays enhancing the detection of IFD-specific antigens, recent advances combining conventional fungal antigen assays and clinical investigations has drastically increased diagnosis accuracy [65,66]. The next development in diagnostic capability will likely involve the identification of host biomarkers that predict patients highly susceptible to developing IFD or even identify patients with IFD. Whilst the fungal species and their infection mechanisms that cause IFD are highly diverse and thus challenging to diagnose, changes in key immune components that modulate anti-fungal immunity may represent key host biomarkers of IFD. Crucially, antifungal immunity relies heavily on C-type lectin receptors and the subsequent immune response underpinning host biomarkers may broadly apply to multiple different pathogenic fungal species and types of IFD, and thus multiple different patient groups. If identified early enough these host biomarkers may even predict IFD prior to fungal biomarker detection. The rest of the review will discuss the recent advances in the identification and utilisation of host biomarkers that, in combination with existing methods could optimise IFD management and drive a personalised medicine approach to IFD risk analysis, diagnosis and treatment.

## 7. Single Nucleotide Polymorphism Host Biomarkers

Of the recent advancements in IFD diagnosis, perhaps the most well defined are host genetic risk factors associated with an increased incidence of IFD. Crucially, any genetic biomarkers associated with increased IFD susceptibility could be identified prior to immune suppressive treatment and subsequent risk of IFD; thereby, permitting a truly pre-emptive and personalised approach to each patient’s fungal investigations and potential antifungal strategies. It appears likely that genetic markers that predispose to IFD and demonstrate increased risk of developing IFD, will unlikely provide a sole diagnostic role but could be combined with other tests to generate a composite probability of IFD. With the mainstream adoption of next-generation sequencing platforms and our increasing understanding of the human genome, screening patients for mutations or deficiencies that mediate IFD susceptibility would not be technically challenging. This process could also be applied to stem-cell transplant donors to limit the introduction of genes containing mutations associated with increased fungal susceptibility in already at-risk transplant recipients, but this would likely be offset by HLA matching and already established risk factors such as CMV status.

The first major host genetic biomarker associated with IFD was caspase recruitment domain 9 (*CARD9*) mutation. *CARD9* is an adaptor molecule that transduces C-type Lectin-like Receptor (CLR) signalling. CLRs are a subset of pattern recognition receptors that primarily recognise fungal pathogens and induce largely protective immune responses. Patients with *CARD9* mutations leading to CLR signalling deficiency were susceptible to IFD without any immune suppression [67,68]. Upstream of *CARD9*, mutations in the CLR Dectin-1 have been associated with IFD. Dectin-1 is a CLR that binds (1-3)-β-D-glucan located on the fungal cell wall of *Aspergillus*, *Candida*, and numerous other fungi that cause human disease. Multiple studies have identified the central role of Dectin-1 mediating protective antifungal immunity [69,70,71,72]. The Y238X mutation in Dectin-1 results in a truncated receptor, poor expression and reduced signalling and was first associated with recurrent mucocutaneous fungal infections such as vulvovaginal candidiasis [73], before being associated with an increased IA incidence [74]. Haematology patients with the Y238X mutation are highly susceptible to IA due to reduced fungal recognition and immune responses [74,75]. To date, no study has revealed an increased incidence of Dectin-1 Y238X in non-*Aspergillus* IFD despite Dectin-1 mediating antifungal immunity against many fungal species.

In addition to Dectin-1 mutations, mutations in the CLRs DC-SIGN and Dectin-2 have been associated with IFD incidence. Multiple single nucleotide polymorphisms (SNPs) in DC-SIGN that impact the CLR’s RNA expression have been associated with increasing IA incidence in haematology patients [76,77]. A SNP in the fungal-binding CLR Dectin-2 was recently identified in a haematology patient who developed IA. The Dectin-2 N170I mutation resulted in a frame shift and early stop codon, truncating the receptor which was subsequently not correctly assembled and expressed. Peripheral blood mononuclear cells (PBMCs) from the Dectin-2 N170I patient displayed reduced inflammatory responses against *Aspergillus.* Additionally, Dectin-2 deficient cells produce reduced inflammatory responses against *Aspergillus* and *Candida*, suggesting that this mutation may act as a biomarker to indicate those more susceptible to IC as well as IA [78].

Aside from SNPs resulting in CLR deficiencies, mutations that may increase IFD susceptibility have been found in numerous innate and adaptive immune components and pathways, such as the Pentraxin-3 gene. In 2014, genetic variants of the associated gene were linked to an increased risk of IA in allogeneic SCT patients [79]. Since this initial study, the mechanism by which these genetic variants mediate susceptibility through reduced Pentraxin-3-dependent *Aspergillus* opsonisation in the lung and subsequently reduced neutrophilic killing and inflammatory response has been well defined [80]. Crucially, several additional studies have validated the clinical significance of Pentraxin-3 genetics in multiple patient cohorts providing clear evidence that genetic variants of Pentraxin-3 mediate fungal susceptibility and should be used as a host biomarker of fungal susceptibility [81,82,83,84].

A larger study in 2015 investigated 36 SNPs within 14 immune-modulating genes as biomarkers for IA in a cohort of at-risk cancer patients. This study identified target genes through criteria including that SNPs have laboratory evidence of biological function and/or have been associated with other infectious diseases. A total of 781 patients (149 IA cases) were screened and significant associations between SNPs in IL-4R and IL-8 were associated with an increased risk of IA. This risk increased further in allogeneic SCT patients which the authors propose is the result of prolonged and severe immune suppression that enhances the effects of these mutations on IA susceptibility. Interestingly, gene variants in IL-12B and IFNγ decreased IA risk suggesting these host biomarkers may be used as exclusion criteria for IFD. Importantly the mutations found to mediate IA risk were characterised in vitro. For example, the IFNγ variant increased macrophage-mediated fungal conidia neutralisation [85].

Whilst some gene variants in key anti-fungal immune pathways have been well defined, many other SNPs require functional characterisation before their full application into diagnostics may be achieved. Here, amongst the extensive list of SNPs associated with fungal disease incidence, mutations in TLRs and major cytokines (IL-1β and IL-6) that convey broad protection against fungal pathogens may most easily be applicable as diagnostic biomarkers in clinical settings [86,87,88]. Many TLR SNPs have been found in patients with fungal disease, but few have been significantly associated and characterised to increase IFD susceptibility [87,88,89]. These key PRRs and cytokines are convenient to investigate due to the large repertoire of in vitro and in vivo models of these immune components. The vast majority of SNP-association studies have focused on increased IFD susceptibility, it may also be interesting to consider any gene variants associated with reduced IFD susceptibility and how these may be used as exclusion criteria in the clinic.

Identifying, characterising and utilising SNPs as biomarkers associated with pan-fungal susceptibility has huge potential to improve management and target anti-fungal therapies based on risk stratification. However, there are limitations that must be overcome. Currently, each SNPs relevance to a specific fungal pathogen/IFD/patient cohort restricts their usefulness as broad prospective biomarkers for IFD. The studies that have investigated genetic variants in populations at risk of IFD have been limited in size and scope, and often focus on one patient group and one type of IFD, which does not necessarily reflect the clinical risk. It is likely that the widespread next generation sequencing of populations at risk of IFD will be required to overcome sample size issues and population heterogeneity. This itself presents a huge challenge, as this technology is currently expensive and requires skilled users and extensive analysis. Until an automated sequencing pipeline with a defined protocol is achieved, smaller study results with manual patient sequencing will continue to move the field forwards. It is also highly likely that many new genetic variants associated with increased IFD incidence will be identified through next generation sequencing. Characterising and analysing the impact of these mutants will be a large undertaking. Critically, how genetic variants in patients are used to inform clinical decisions requires extensive investigation.

## 8. Gene Expression Host Biomarkers

Alongside advances in the understanding of genetic mutations and IFD, recent studies have investigated patient’s gene expression of key antifungal immune components and the subsequent susceptibility to IFD. Given SNPs located in anti-fungal immune receptors such as Dectin-1 and Dectin-2 result in reduced receptor expression, reduced signalling and increased IFD risk [74,78], non-SNP-associated reduced expression would likely also result in diminished signalling and increased fungal susceptibility. In a recent study in haematology patients, individual gene expression of the CLRs *Dectin-1*, *Dectin-2*, *Mincle* and *Mcl* were quantified by RT-qPCR (prior to the initiation of their primary haematology disease treatment) and associated with the incidence of IA. Whilst the results lacked power due to sample size, patient groups with lower Dectin-1 expression were 10 times more likely to develop IA whilst patient groups with higher Mcl expression appeared less likely to develop IA. Interestingly, when results were combined with other biomarkers a predictive IFD incidence model was generated with promising retrospective conclusions that merit further investigation [90].

Expression changes in microRNAs have also been explored as host biomarkers for IFD. A pilot clinical study in haematology patients investigated the diagnostic utility of 14 microRNAs and identified 4 (miR-142-3p, miR-142-5p, miR-26b-5p and miR-21-5p) that were significantly overexpressed in patients with IA. These results were generated from RT qPCR assays from patient blood which is within current clinical diagnostic capabilities [91]. Whilst not validated in a clinical setting, two further micro RNAs (miR-132 and miR-155) were induced in immune cells following *Aspergillus* challenge, providing more evidence that microRNAs induced by fungi may be useful biomarkers of IFD, and given these are generated by immune cells in response to the presence of *Aspergillus* within the host, could play a role in a pre-emptive diagnostic capacity [92]. However, the functional role these microRNAs convey over immunity and disease progression requires clarification.

Another layer of immune mediation arises from CLR splicing. The expression of Dectin-1 isoforms following alternative splicing, particularly isoforms that lack exon 3 (and therefore a stalk region), mediated susceptibility to fungal infection [72,93]. Dectin-2 also has alternative spliced isoforms although their relevance to immunity is not clear [94]. The topic of CLR alternative splicing and its impact on antifungal immunity requires thorough investigation before being applied to clinical diagnostics. Once better understood, isoforms of receptors could be relatively easily identified through mRNA RT-qPCR on genetic material routinely extracted from patients.

Investigating a patient’s gene expression, micro RNA expression and splicing of key anti-fungal components may greatly enhance our understanding of why only some at-risk patients develop IFD. The incorporation of RNA sequencing technologies, particularly single-cell RNA sequencing, may reveal numerous gene expression host biomarkers that mediate IFD susceptibility. However, these technologies are very labour intensive, costly and generate large bioinformatic databases that require extensive analysis. Until these protocols could be streamlined and/or automated, the adaption of these technologies into clinical settings will be difficult. Further challenges to overcome include the timing and type of sample taken from patients at risk of IFD. Gene expression is often cell specific and would likely change throughout patient’s treatment for their primary disease and/or IFD.

## 9. Cytokines and Chemokines as Host Biomarkers

Recent advances in host biomarkers for IFD are not limited to gene mutations and expression variation. Studies have started to investigate host cytokines/chemokines as IFD biomarkers, and utilised host-derived functional assays with cytokine/chemokines as readouts to predict or identify IFD. A recent study identified that treatment with the TNF blocker etanercept after allogeneic stem cell transplant (SCT) was associated with increased risk of developing IA. Investigation into the mechanism of increased susceptibility through RNAseq analysis revealed that treatment of *A. fumigatus* stimulated monocyte-derived macrophages with etanercept resulted in the reduction of genes downstream of TNF (*RELB*, *ICAM1*, *BIRC2*, *CXCL10* and *BCL3*). The authors showed that serum CXCL10 levels were significantly reduced in IA patients under etanercept treatment compared to IA patients without etanercept treatment. This study indicates that etanercept treatment is a risk factor for the development of IA and reduced CXCL10 serum levels are associated with etanercept treatment; however, host CXCL10 is not necessary as a biomarker in this situation as etanercept usage is sufficient as the risk factor but highlights the potential influence of clinical intervention on host response. Reduced TNF and CXCL10 in patients treated with etanercept would hamper a robust defence against *A. fumigatus* [95]. In agreement with this, SNPs in *CXCL10* were associated with increased risk of developing IA in allogeneic SCT patients. Following stimulation with *A. fumigatus*, immature dendritic cells from patients with the *CXCL10* SNPs displayed reduced CXCL10 expression compared to the wild type allele [96].

A recent study has also investigated host serum levels of cytokines/chemokines in the context of IFD. The transcriptomic analysis of samples from three probable IA cases and three control patients in combination with analysis of six in vitro studies resulted in the identification of 9 targets (*MMP1*, *MMP9*, *LGALS2*, *ITGB3*, *VEGFA*, *CASP3*, *CD40*, *CXCL8* and *PAI-1*) for further study. Reduced expression of *MMP-1*, *MMP-9* and *ITGB3* RNA levels and increased expression of *LGALS2* RNA levels in IPA patients versus controls were confirmed by qPCR. Furthermore, serum levels of IL-8 and Caspase 3 protein were increased in probable IPA patients in two centres, and serum levels of MMP-1 protein were increased in probable IPA patients in one centre only. The authors suggested that *LGALS2* and *MMP1* RNA levels and serum IL-8 and caspase-3 levels could be used as potential host biomarkers in combination with traditional fungal biomarkers to enhance the diagnosis of IA in patients post-SCT. Interestingly, the same study found that CAPA patients had lower serum IL-8 and Caspase 3 protein levels compared to COVID-19 patient controls, possibly highlighting a risk factor for infection, rather than a diagnostic aid in this cohort, indicating that a different host biomarker strategy may be required for different patient cohorts [97].

Another approach recently undertaken utilised patient cells *ex-vivo* to investigate cytokine and chemokine responses that may be used as host biomarkers for IFD. A small study in haematology patients designed an *ex-vivo* assay where PBMCs were stimulated with LPS or *Aspergillus* and their cytokine responses against each challenge quantified. The study retrospectively showed that 62.5% of SCT patients developed IA, when their PBMCs did not produce IL-6 or TNF in response to ex vivo stimulation with *A. fumigatus* while they produced IL-6 or TNF in response to LPS. The incidence of IA in a subset of SCT patients increased to 80% when IL-6/TNF production in response to *A. fumigatus* and LPS was combined with expression levels of Dectin-1 and Mcl as discussed in the previous section [90]. This study indicates that IL-6/TNF levels in response to *A. fumigatus* and LPS have the potential to be used as host biomarkers to demonstrate increased risk of IA but possibly aid diagnosis when used in combination with other host factors and mycological evidence.

Finally, a whole blood assay was recently developed to measure the release of T cell signature cytokines in response to *A. fumigatus* antigens. Whole blood was co-stimulated with α-CD28 and α-CD49d in combination with *A. fumigatus* lysate. Patients with *A. fumigatus* associated lung pathologies displayed a trend towards increased T helper signature cytokine responses (IL-4, IL-5, IFNγ, IL-17 and IL-13) in the assay compared to control patients with other chronic lung diseases. IL-4 and IL-5 levels were significantly increased in the *Aspergillus*-associated lung disease patients, while the other cytokines did not reach statistical significance. However, all the T cell signature cytokines displayed non-overlapping interquartile ranges of concentrations between *Aspergillus*-associated lung disease patient samples and control patient samples, possibly limiting its role as diagnostic test, unless used in combination with mycological testing. While this assay requires further testing and validation in a larger patient cohort, it shows promise as an immune surveillance assay for patients with opportunistic *A. fumigatus* infections or mould reactive hypersensitivity syndromes [98].

These studies clearly show that cytokines and chemokines have compelling potential to be used as host biomarkers for IFD. In agreement, diagnostic assays that use cytokines as biomarkers are being widely investigated in the infectious disease field [99,100,101]. Whether inflammatory responses following ex vivo fungal stimulation or cytokine/chemokine levels in serum, quantification of these host molecules can be used to identify IFD susceptibility/risk. However, this area of research is newly being applied to complicated immune compromised patients at risk of IFD and will eventually need to overcome several challenges before informing clinical decisions. Whilst the identification of serum cytokines and chemokines could be relatively easily achieved in clinical settings, the PBMC ex vivo assay would need to be streamlined as extraction and purification of PBMCs and then the *ex-vivo* assay would be labour intensive and time consuming. The whole blood assay has the advantage of not having to extract PBMCs; however, further testing and validation is still required before it could be used in a clinical setting.

## 10. Host Biomarkers in IFD Predictive Modelling

Recently, there have been great efforts to identify host-derived biomarkers that mediate susceptibility to IFD (Table 2). Importantly, these biomarkers on their own have been shown to identify patients at particularly high risk of IFD. However, their most compelling use is in collaboration with conventional fungal biomarkers and clinical risk of IFD.

Utilising CLR SNPs as IFD biomarkers alongside conventional biomarkers has been investigated with excellent results even in relatively small patient cohorts. The presence of DC-SIGN and Dectin-1 mutations was used in a predictive model with clinical risk factors (e.g., allogeneic SCT, respiratory virus infection) to identify high-risk patients. The presence of two genetic host biomarkers increased the IA risk to 4.7%, far higher than the 0.6% IA risk in patients with no biomarkers [65]. SNPs were also successfully used as biomarkers to predict patients who would develop IA in a larger multi-centre study. The predictive model used in this study included the use of 4 gene variants as host biomarkers alongside conventional risk factors including age, gender, allogeneic SCT and antifungal prophylaxis. The results from this study identified a significant improvement in predicting IA when the SNP biomarkers were included alongside clinical risk and fungal biomarker in the prediction model compared to just using clinical risk and fungal biomarkers, highlighting how host biomarkers could be used in a diagnostic strategy [85].

A smaller retrospective study combined genetic and protein (cytokine) host biomarkers with conventional clinical and fungal biomarkers to devise a predictive strategy for IFD risk and a possible aid to diagnosis. This study defined high risk patients as those with both high gene expression of the CLRs Dectin-1 and Mcl, and an LPS-induced TNF/IL-6 response from *ex-vivo* stimulation of PBMCs but a lack of *Aspergillus* TNF/IL-6 response. Of the patients that possessed these host biomarkers 80% (4 out of 5) developed IA. Interestingly, patients that produced a TNF/IL-6 response following ex vivo PBMC stimulation did not develop IA. Suggesting this biomarker may reduce or even exclude patient’s IA risk [90]. The results warrant further investigation primarily examining these host biomarkers in larger patient cohorts and combining them with conventional fungal and clinical biomarkers to enhance the predictive model.

## 11. Conclusions

The significant clinical importance of IFD is clear and the number of IFD cases is increasing. Previously, the diagnosis of these infections has predominately used clinical risk factors and mycology to inform clinical decisions and regularly relies on empirical antifungal therapy. However, the causative fungi behind IFD are highly diverse, frequently encountered and/or commensals, and drive IFD through a wide range of infectious mechanisms. The diagnostic challenges IFD fungi pose has resulted in diagnostic strategies optimal for excluding IFD, with limited specificity for targeting anti-fungal therapy. Recent investigations into IFD diagnosis have focused on host-derived biomarkers that can be used to predict or even identify IFD. Here, novel genetic host biomarkers such as SNPs in key anti-fungal immune molecules, novel gene expression host biomarkers, and novel protein host biomarkers such as serum cytokines or cytokine responses following ex vivo fungal stimulation have been associated with IFD. Furthermore, these host biomarkers have been used alongside clinical and diagnostic biomarkers to successfully predict IFD risk and possibly aid diagnosis in patients. Whilst many of these studies require further development in larger patient cohorts, they begin to describe host biomarkers that, when incorporated alongside conventional diagnostic strategies, may greatly improve IFD diagnosis, predict high-risk patients, and allow a personalised medicine approach to target-fungal therapies (Figure 1). Currently, it remains unclear as to whether individual host biomarkers could determine an increased risk of IFD or be indicative of actual IFD. To assess this, large scale prospective studies of promising host biomarkers must be undertaken. These studies should also consider the baseline levels of any host biomarkers without IFD to fully clarify their role. For instance, if a host biomarker is typically absent in non-IFD cohorts, then its presence in a patient will likely be indicative of increased risk and possibly provide a degree of diagnosis. However, diagnosis will likely require combining with clinical/radiological evidence or mycology typical of the specific IFD. Given that the current diagnosis of IFD is based on various degrees of certainty (probable or possible IFD), with proven IFD less encountered, host biomarkers associated with increased risk or probability of IFD are well placed to complement or enhance this approach.

## Figures and Tables

**Figure 1 jof-08-01307-f001:**
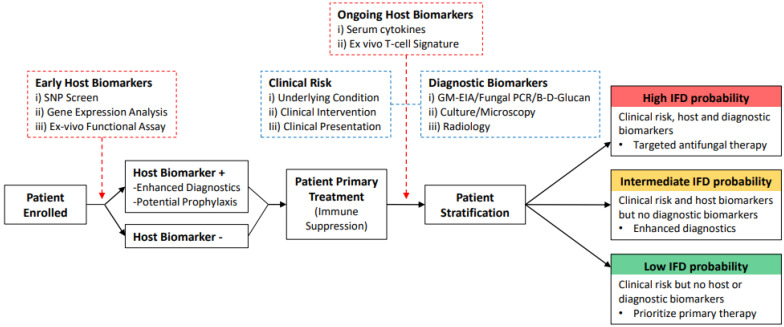
**Incorporating host biomarkers into IFD diagnostic strategies.** Prospective IFD diagnostic strategy that utilizes the latest advancements in IFD-associated host biomarkers alongside conventional fungal biomarkers and clinical risk to better target anti-fungal prophylaxis [65,85,90,96,97,98]. Blue dotted boxes represent current investigations into IFD susceptibility and incidence. Red dotted boxes represent potential new host biomarker investigations into IFD susceptibility and incidence. Solid boxes represent patient pathway and potential stratification into high, intermediate, and low IFD risk cohorts.

**Table 1 jof-08-01307-t001:** The performance of molecular diagnostics and biomarker assays for the diagnosis of IFD.

Assay	Sample Type	Performance Parameter					Reference
		Sensitivity (%)	Specificity (%)	LR + Tive	LR − Tive	DOR	
Candida PCR	Blood	95.0	92.0	11.88	0.05	218.50	[36]
T2 Candida	Blood	91.0	94.0	15.17	0.09	158.41	[37]
Aspergillus PCR	Blood	88.0	75.0	3.52	0.16	22.00	[38]
		84.0	76.0	3.50	0.21	16.60	[39]
		79.2	79.6	3.88	0.26	14.86	[40]
	BAL fluid	78.4	93.7	12.44	0.23	53.98	[41]
		79.6	94.1	13.49	0.22	62.23	[42]
		76.8	94.5	13.96	0.25	56.88	[43]
PCP PCR	Respiratory	97.0	94.0	16.17	0.03	506.56	[44]
		98.0	91.0	10.89	0.02	495.44	[45]
		99.0	90.0	9.90	0.01	891.00	[46]
GM EIA	Blood	79.3	80.5	4.07	0.26	15.81	[47]
		79.3	86.3	5.79	0.24	24.13	[48]
	BAL fluid	83.6	89.4	7.88	0.18	42.99	[49]
		85.7	89.0	7.79	0.16	48.49	[50]
		92.0	98.0	46.0	0.08	563.5	[51]
Β-D-Glucan	Blood	76.8	85.3	5.22	0.27	19.21	[52]
		78.0	81.0	4.11	0.27	15.11	[53]
	BAL fluid	52.0	58.0	1.24	0.83	1.50	[54]
*Aspergillus* LFA	Blood	68.0	87.0	5.23	0.37	14.22	[55]
	BAL fluid	86.0	93.0	12.29	0.15	81.6	
Cryptoccocal LFA	Blood	97.9	89.5	9.32	0.02	397.37	[56]
	CSF	99.5	99.5	199	0.01	39601	

Key: LR + tive: Positive likelihood ratio; LR − tive: Negative likelihood ratio; DOR: Diagnostic odds ratio; BAL: Bronchoalveolar lavage fluid; LFA: Lateral flow assay; CSF: Cerebrospinal fluid.

**Table 2 jof-08-01307-t002:** IFD Host biomarkers for IFD that have been identified and validated in clinical studies.

Biomarker Category	Host Biomarker	Reference	IFD Susceptibility
SNPs	*CARD9*	[67,68]	↑
	*DC-SIGN (CD209)*	[76,77]	↑
	*CLEC7A* (Dectin-1)	[73,75]	↑
	*CLEC6A* (Dectin-2)	[78]	↑
	*PTX3* (PENTRAXIN-3)	[79,80,81,82,83,84]	↑
	*IL-4R*	[85]	↑
	*CXCL8* (IL-8)	[85]	↑
	*IL-12B*	[85]	↓
	*IFN* *γ*	[85]	↓
	*CXCL10*	[96]	↑
Gene Expression	Reduced *CLEC7A* (Dectin-1)	[90]	↑
	Reduced *CLEC6A* (Dectin-2)	[90]	↑
	Increased *CLEC4D* (Mcl)	[90]	↓
	Increased miR-142-3p	[91]	↑
	Increased miR-142-5p	[91]	↑
	Increased miR-26b-5p	[91]	↑
	Reduced *MMP1*	[97]	↑
	Increased *LGALS2*	[97]	↑
Gene Splicing	Truncated *CLEC7A* (Dectin-1)	[72,93]	↑
Cytokines/Chemokines	Etanercept treatment (TNF blockade)	[95]	↑
	Increased serum IL-8	[97]	↑
	Increased serum Caspase-3	[97]	↑
	ex vivo PBMC fungal stimulation low/absent TNF response	[90]	↑
	ex vivo PBMC fungal stimulation low/absent IL-6 response	[90]	↑
	ex vivo whole blood T-cell assay increased IL-4	[98]	↑
	ex vivo whole blood T-cell assay increased IL-5	[98]	↑

Key: Red up arrows indicate increased IFD risk. Green down arrows indicate reduced IFD risk.

## Data Availability

Not applicable.

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
