# Peer review of "The Use of Host Biomarkers for the Management of Invasive Fungal Disease"

_jof, 2022, doi:10.3390/jof8121307_

Round 1
Reviewer 1 Report
Dear Authors,
I read your manuscript concerning the identification of host biomarkers for IFD, their use in predictive modelling and their potential application/usefulness in informing clinical decisions. The paper is rich in information and presents a comprehensive overview. I report some notes to improve the manuscript.
1) Abstract should be structured
2) For each paragraph, you should summarize the results in tables.
3) The material and method section is missing. The are no inclusion or exclusion criteria based on the research. I suggest that the author include a research strategy.
4) Line 147, include the histology as confirmation of pathology
5) Line 183, flucytosine is no longer used in clinical practice
6) I suggest reading and citing:
- Campione E, Cosio T, Lanna C, Mazzilli S, Ventura A, Dika E, Gaziano R, Dattola A, Candi E, Bianchi L. Predictive role of vitamin A serum concentration in psoriatic patients treated with IL-17 inhibitors to prevent skin and systemic fungal infections. J Pharmacol Sci. 2020 Sep;144(1):52-56. doi: 10.1016/j.jphs.2020.06.003. Epub 2020 Jun 11. PMID: 32565006.
Author Response
We thank the reviewers for taking the time to review our manuscript and providing us with insightful feedback, which have done our best to address and described below:
Reviewer 1
- Abstract should be structured
Response: Following JOF instructions we did not structure the abstract and utilize specific headings. Given this is a review article, the headings typically used for original research are not easily applied. We therefore have not formally structured the abstract and leave this requirement to your discretion.
- For each paragraph, you should summarize the results in tables.
Response: We thank the reviewer for this suggestion and have included table 2 in the manuscript which lists the host biomarker and risk of IFD, and feel this greatly improves the manuscript.
- The material and method section is missing. The are no inclusion or exclusion criteria based on the research. I suggest that the author include a research strategy.
Response: This is a summary review of the topic, it is not a systematic review and/or meta-analysis, as the topic lacks the data to perform such an undertaking and as such search criteria are not available and the topics included and manuscripts cited may not be exhaustive. We have added tentative inclusion criteria as follows “We will then investigate the potential application of new host biomarkers to aid in the management of IFD, through a summary review of publications describing research only validated in clinical cohorts.”
- Line 147, include the histology as confirmation of pathology
Response: We were not fully certain of where the reviewer was alluding to, but felt the changes to lines 132-135 in the current manuscript (Lines 118-120 in the previous version) and lines 167-170 in the current manuscript (lines 147-149 in previous version) covered the point raised.
Lines 132-135 now read “Beyond the prediction of clinical risk, patients are further stratified into proven, probable, and possible IFD based on histological confirmation of pathology or the combination of clinical/radiological evidence and mycology [33].”
Lines 167-170 now read “While combined fungal biomarker strategies have demonstrated utility for excluding IFD and subsequent need to treatment, their use is not widespread, a limitation that can also be applied to the histological investigation of tissue biopsies when trying to confirm IFD [32, 33].”
- Line 183, flucytosine is no longer used in clinical practice
Response: Flucytosine is still used clinically in combination with other antifungals (amphotericin B in particular) for the treatment of cryptococcosis and rare yeast infections. Subsequently, we have left this sentence unchanged. Please note: The only reference to Flucytosine is on line 156 of the original manuscript and line 177 of the revised manuscript, this does not relate to the line number stated by the reviewer
- I suggest reading and citing:
They provide a review reference - https://pubmed.ncbi.nlm.nih.gov/32565006/ and https://gut.bmj.com/content/61/12/1693
Response: We thank the reviewer for suggesting this reference, however, after updating our inclusion criteria to biomarkers identified and validated in clinical research studies, we do not feel we can include this work on vitamin A. Whilst this work is of keen interest to the review topic, and the role of vitamin A and ATRA has been well shown in vitro and in vivo, there is not statistically significant clinical validation of this pathway.
Reviewer 2 Report
The authors propose to perform a review on the latest developments in the identification of host biomarkers for IFDs. The topic is of interest, but there are some issues that should be addressed:
1 - It is important to clarify if the review is focused specifically in opportunistic IFDs: you comment about opportunistic IFDs in the abstract, but it should be stated on the text.
2- In the "overview of the Invasive Fungal disease" topic, it would be important to comment briefly about other complexes of species within the genders Aspergillus and Candida. Furthermore, the topic "Current Fungal Diagnosis and Therapeutics" needs some review, in order to include new data on fungal biomarker and molecular diagnosis performance for IFDs diagnosis.
3 - In terms of the main objective of the paper that is "to summarise the latest developments in the identification of host biomarkers for IFD, their use in predictive modelling and their potential application/usefulness informing clinical decisions", there are interesting information. As the title states "Host Biomarkers for the Diagnosis of Invasive Fungal Disease", a better distinction between host biomarkers that elevate the risk versus wchich of them could be effectively used to improve IFDs diagnosis is important throughout the text.
4- Maybe the term “at-risk” between the commas could be substituted by population at risk or populations at higher risk for developing IFDs.
Author Response
We thank the reviewers for taking the time to review our manuscript and providing us with insightful feedback, which have done our best to address and described below:
Reviewer 2
1 - It is important to clarify if the review is focused specifically in opportunistic IFDs: you comment about opportunistic IFDs in the abstract, but it should be stated on the text.
Response: We have made this clearer by amending the first abbreviation of IFD (line 21)
2- In the "overview of the Invasive Fungal disease" topic, it would be important to comment briefly about other complexes of species within the genders Aspergillus and Candida. Furthermore, the topic "Current Fungal Diagnosis and Therapeutics" needs some review, in order to include new data on fungal biomarker and molecular diagnosis performance for IFDs diagnosis.
Response: We have added further information regarding the species of Aspergillus responsible for disease and highlighted current diagnostic limitations related to the various species in the section “an overview of invasive disease”.
Lines 54-59 now read “Aspergillus fumigatus is primarily responsible for IA but other species including Aspergillus flavus, Aspergillus niger, Aspergillus terreus and Aspergillus nidulans are also responsible for human disease, sub-species with the A. fumigati complex are difficult to differentiate from A. fumigatus but may have markedly different antifungal susceptibility profiles [6-9]”
Lines 71-76 now read “Current diagnostic investigations and clinical intervention guidelines (as defined by the EORTC) are limited by their inability to discriminate between different Aspergillus spp. with diagnoses regular achieved in the absence of culture or species level identification [17]. The performance of assays can vary dependent on the species, with many assays designed/optimized for the detection of the most prevalent species (i.e. A. fumigatus)
The main causes of invasive candidal disease were already stated in the text (Lines 95-96)
We have also included some further text on the performance of current biomarkers and molecular tests for IFD (Lines 139-143, Line 146, Line 148) and to keep the word count to a minimum we have included table 1 summarizing the performance of various diagnostic tests.
3 - In terms of the main objective of the paper that is "to summarise the latest developments in the identification of host biomarkers for IFD, their use in predictive modelling and their potential application/usefulness informing clinical decisions", there are interesting information. As the title states "Host Biomarkers for the Diagnosis of Invasive Fungal Disease", a better distinction between host biomarkers that elevate the risk versus wchich of them could be effectively used to improve IFDs diagnosis is important throughout the text.
Response: We have modified the title to reflect this point which now reads “The use of Host Biomarkers for the management of Invasive Fungal Disease”. We have clarified host biomarkers and effect of IFD susceptibility in a new table 1 included in the manuscript.
We have attempted to highlight the point raised whenever possible throughout the manuscript (Lines 203-206, Lines 214-216, Lines 285-287, Lines 322-37, Lines 380-383, Lines 393-395, Lines 403-406, Lines 438-441, Lines 442-444, Lines 466-468)
However, we feel given the preliminary nature of this field it would be too early to state which host biomarkers could be used to demonstrate risk or achieve a diagnosis and have made this clear in the conclusion and lines 472-482 now read “Currently, it remains unclear as to whether individual host biomarkers could determine an increased risk of IFD or be indicative of actual IFD. To assess this, large scale prospective studies of promising host biomarkers must be undertaken. These studies should also consider the baseline levels of any host biomarkers without IFD to fully clarify their role. For instance, if a host biomarker is typically absent in non-IFD cohorts, then its presence in a patient will likely be indicative of increased risk and possibly provide a degree of diagnosis. However, diagnosis will likely require combining with clinical/radiological evidence or mycology typical of the specific IFD. Given that the current diagnosis of IFD is based on various degrees of certainty (probable or possible IFD), with proven IFD less encountered, host biomarkers associated with increased risk or probability of IFD are well placed to complement or enhance this approach.”
4- Maybe the term “at-risk” between the commas could be substituted by population at risk or populations at higher risk for developing IFDs.
Response: This has been updated throughout to improve readability as suggested.